# Experimental analysis of the stick-slip characteristics of faults at different loading rates

**Bao-xin Jia[1,2]\*, Zong-xian Gao[1], Xiu-hui Han[3], Jia-xu Jin[1], Jian-jun Zhang[1]**

**1** School of Civil Engineering, Liaoning Technical University, Fuxin, Liaoling, China, **2** Institute of Geology, China Earthquake Administration, Beijing, China, **3** College of Resources & Safety Engineering, China University of Mining & Technology, Beijing, China

\* jbx_811010@126.com

**Data Availability Statement:** All relevant data are within the paper.

**Funding:** This work was supported by The National Natural Science Foundation of China (No. 51774173), the Natural Science Foundation of

## Abstract

In deep underground engineering, in a large spatial, high-stress environment, rapid excavation is likely to affect the loading rate of the fault structure and to cause stick-slip. In this study, an experiment was conducted to explore the stick-slip characteristics at different loading rates. A double-sided shear experiment and the digital speckle correlation method were used to analyze the evolution of the displacement field, the slip displacement, and the slip rate of the fault's stick-slip activity at different loading rates as well as their correlation with the loading rate. The loading rate, moment magnitude, and stress drop of the fault's stick-slip and their corresponding relationships were studied. The results show that the occurrence of stick-slip is inversely proportional to the loading rate. The evolution of the fault-slip displacement field at different loading rates is similar. At a given loading rate, the magnitude is positively correlated with the stress drop. The magnitude and stress drop are inversely related to the loading rate.

## Introduction

As basic and common geological structures, faults are widely found in underground engineering. The slippage of faults is a common phenomenon of rock friction and can be divided into two different phenomena: smooth and sticky. Fault-slips are characterized by a two-disc lock-out of the fault, and the stress continuously accumulates. When the stress reaches or exceeds the friction between the two plates, the fault suddenly slides, producing relative displacement and relieving stress. During recent years, as China's geotechnical engineering construction, e.g., mining resources, underground storage facilities, water conservancy, and hydropower, has gradually developed in deep underground and high stress areas, accidents such as roof collapse, landslides, water inrush, and impact ground pressure caused by engineering-induced fault slips have increased. Therefore, in-depth research on fault slip characteristics is of important practical significance for the prediction, prevention, and control of disasters in geotechnical engineering [1,2,3,4,5]

Fault slips are one of the hot topics worldwide in the field of seismic and geotechnical engineering, and numerous studies have been carried out. [6] conducted stick-slip experiments

Liaoning Province (No. 201602351), and the Open Fund for the State Key Laboratory of Seismological Dynamics (No. LED2015B01) to BJ. The funders had no role in study design, data collection and analysis, decision to publish, or preparation of the manuscript.

**Competing interests:** The authors have declared that no competing interests exist.

**Abbreviations:** G, The shear modulus of the rock; D, The stick-slip displacement; A, The area of the slip surface of the fault; $M_0$, The seismic moment; $M_w$, The moment magnitude; $\Delta\sigma$, The actual stress drop; $\Delta\sigma_V$, The total stress drop; T, The time of sudden displacement change during stick-slip; $t_V$, The time of the actual stress drop.

based on three structural models to explore the relationship between the type of stick-slip, the stress drop, and the magnitude. [7] studied the entire process of fault evolution from stable sliding to stick-slip. [8] explored the slow fault-slip mechanism through laboratory observations. In addition, [9] conducted a double-sided shear experiment to analyze the evolution characteristics of fault displacement during inter-slip periods and stick-slip periods. [10] studied the dynamic response of geological materials to acceleration, velocity, and displacement during fracturing and slippage under laboratory conditions. Furthermore, [11] established a discrete element model to simulate a single fault biaxial sliding test and used it to investigate the effects of rock particle elasticity, friction coefficient, fault stiffness, and normal stress on fault evolution. [12] conducted a dynamic direct shear test to study the effects of surface roughness and normal stress on the rock joint friction coefficient and micro-shear deformation mechanism. Through a similarity model experiment, [13] explored the movement law of the overburden strata under mining influences and the evolution of fault stress and analyzed the mechanism of fault reactivation caused by coal seam mining. In terms of theoretical research, based on gradient-dependent plasticity, [14] obtained the theoretical expression for the relative displacement of the fault zone along the observed trend. Based on a mechanical model of the stick-slip of a coal body relative to the top and bottom plates, [15] investigated the conditions of coal stick-slip and established a dynamic equation for the coal body after the occurrence of stick-slip. [16] studied the movement law of the double-slider system under different coupling stiffness conditions, obtaining the sliding mode of the slider system under strong and weak coupling conditions. In terms of numerical computations, [17] used the Lagrangian element method to study the influence of the loading rate on the stress level, the plastic zone size, and the snap-back characteristics of the fault band and elastic rock system. In addition, [18] used the Coulomb shear model to study the spatial and temporal evolutions of the normal stress and shear stress on the fault contact surface in the mining process as well as the movement law of the upper wall and footwall of the faults. According to micro-seismic mine data, [19] performed a back analysis of fault slip data using a three-dimensional numerical simulation of the entire well, obtaining the physical and mechanical properties of the shear band.

Although significant achievements have been made in fault slip research, deep underground engineering construction involving high-stress environments, the exploitation of large spaces, and rapid construction has been increasing in recent years. First, fault stick-slip is likely to occur in a high-stress environment. Second, exploitation of large spaces and rapid construction may cause a loading rate effect of the fault structure. Therefore, with respect to the influence of the loading rate on fault stick-slip, considerable and intense experimental research still needs to be performed to provide further experimental support for establishing theoretical models and for conducting numerical simulation of geotechnical engineering projects influenced by fault structures in deep underground or high-stress environments.

In this study, we use digital image correlation methods to accurately measure the deformation and displacement of rock specimens [20,21,22]. A charge-coupled device (CCD) camera and a high-speed camera were used to build an image acquisition system. Through analysis of the speckle field on the surface of the collected specimens using the digital speckle correlation method, a double-sided shear experiment was carried out to study the evolution characteristics of the displacement field, the slip displacement, the loading curve, the slip rate, the magnitude, and the stress drop of fault stick-slip at different loading rates.

## Materials and methods

In this study, a double-sided shear experiment was conducted to explore the evolutionary characteristics of the fault stick-slip at different loading rates. Fig 1 shows the dimensions,

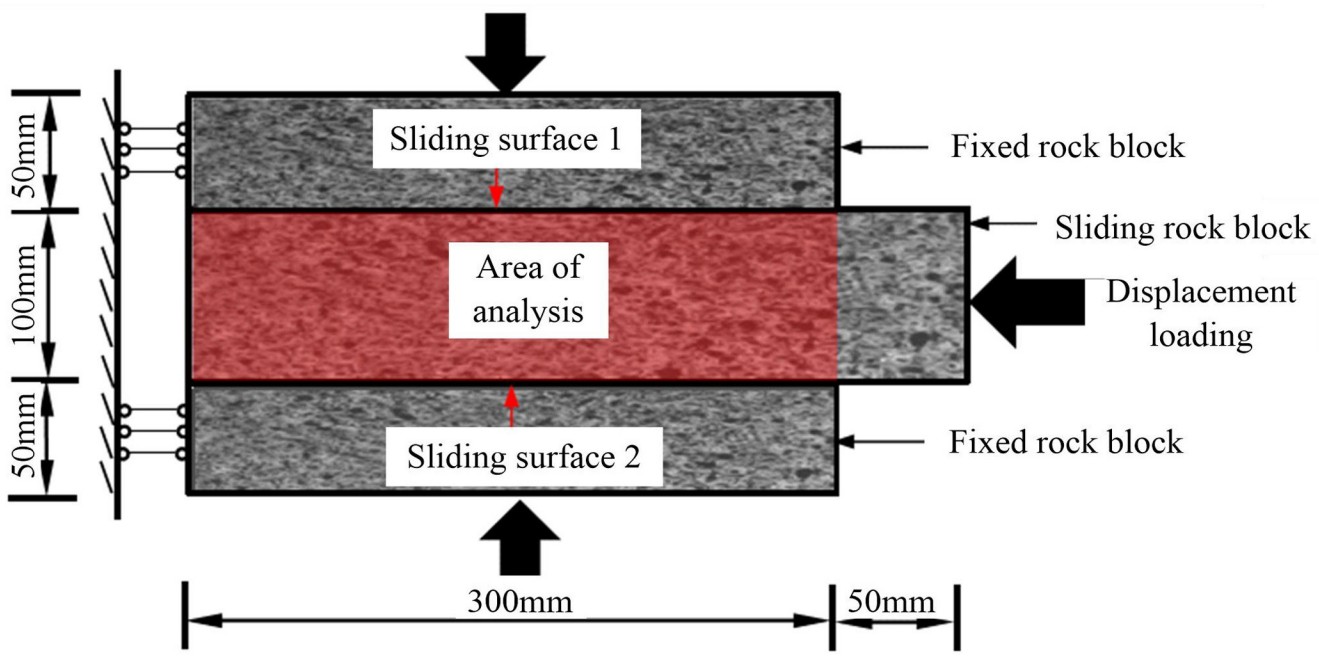

**Fig 1. Experimental model and specimen size.**

displacement boundaries, and loading conditions of the specimens. The rock material was granite with a shear modulus of approximately 25.7 GPa. The specimens used in the fault stick-slip experiment were three pieces of granite, including two fixed rock blocks and one sliding rock block. The dimensions of the fixed rocks was 300 mm × 50 mm × 50 mm (L × W × H), and those of the sliding block was 350 mm × 50 mm × 100 mm. The specimens consisted of two sliding surfaces (Sliding surface 1 and Sliding surface 2), the area of which was 0.015 m$^2$ (300 mm × 50 mm). The sliding surfaces were polished using 300# emery paper, and the artificial speckle field was sprayed on the observation surface of the model.

The displacement boundary and loading conditions of the specimens were as follows. The horizontal displacement on the left side of the two fixed rocks was restricted, and the horizontal displacement on the right side was kept in a free state. Considering the actual stress state of the fault and the compressive strength of the rock specimen and based on the setting of the normal stress used in previous shear tests [23,24], a constant vertical pressure of 25 MPa was selected to create stick-slip conditions. The left side of the sliding block was kept in a free state, and the right side was loaded using the displacement control loading method. According to the variation range of the displacement of the mining project, 18 experiments with 6 loading rates were designed and conducted (three experiments for each loading rate). The loading rates were 4 μm/s, 6 μm/s, 8 μm/s, 10 μm/s, 15 μm/s, and 20 μm/s.

The experimental system consisted of two parts, i.e., the loading device and the image acquisition system. Fig 2 shows the setup of the experimental system. The experimental loading device was a RLJW-2000 hydraulic servo testing machine with a maximum axial load of 3000 kN and a maximum horizontal load of 1500 kN. The data recording frequency range of the loading device was $1 \times 10^{-5}$ to 30 Hz; and the strain loading rate was $1 \times 10^{-7}$ to $1 \times 10^{-2}$ mm/s. The image acquisition system included a CCD camera, a high-speed camera, and a high-speed camera trigger device. CCD is the most commonly used image sensor for machine recording. In this experiment, the model of the CCD camera used was acA1600-20gm. The

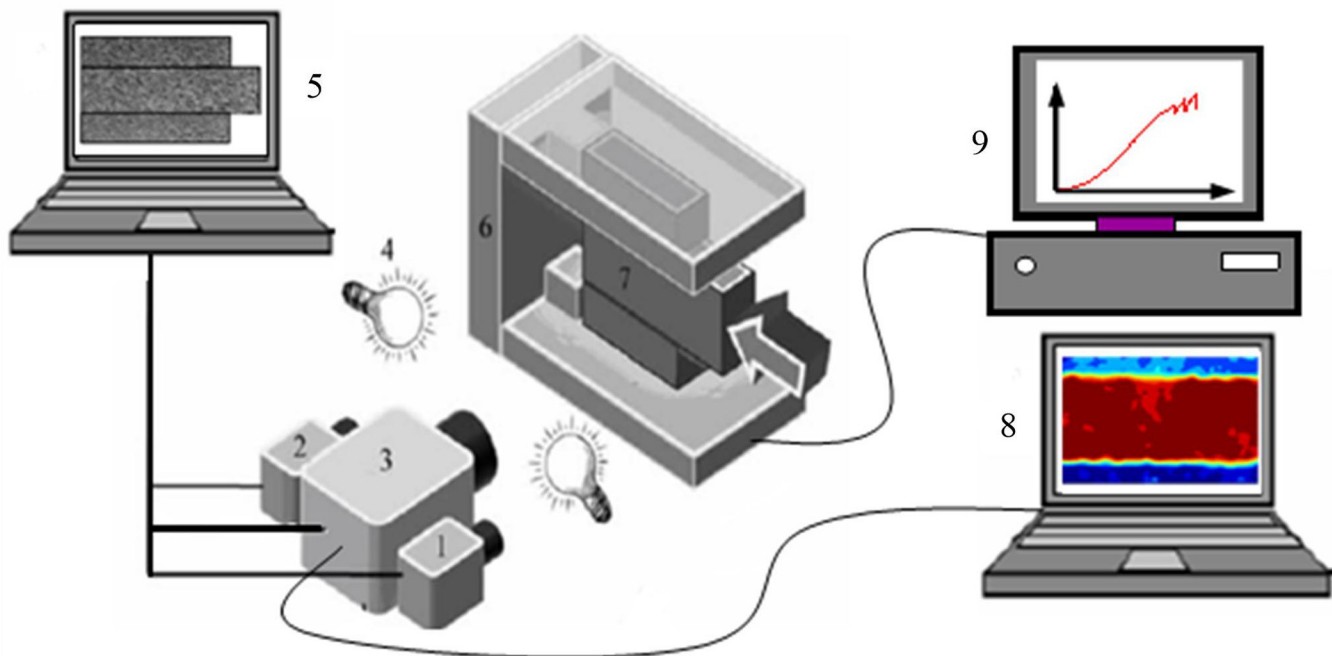

1- CCD camera; 2- CCD camera; 3- High-speed camera; 4- Light source;
5- Computer; 6- Loading device; 7- Specimen; 8- Computer; 9- Computer.

**Fig 2. Schematic of the test system.**

CCD camera was used to collect the speckle images on the surface of the specimens throughout the process, with an acquisition rate of 5 frames per second and an object surface resolution of 0.20 mm per pixel. The high-speed camera was an industrial digital camera, which converted digital image capture targets into image signals and sent them to a dedicated image processing system. The image system performed various operations on these signals to extract the features of the targets. The high-speed camera model used in this experiment was a Photron Fast Cam SA1.1. The high-speed camera was responsible for collecting images of the surface of the specimens during the transient process of the fault stick-slip motion. With the high-decibel sound emitted by the fault stick-slip as the triggering condition, the post-trigger mode was adopted with an acquisition rate of 4000 frames per second and an object surface resolution of 0.50 mm per pixel.

Prior to the experiments, the time of the loading device and that of the image acquisition system were synchronized, and the image acquisition system was debugged. During the experiments, a vertical load was applied to the specimens and remained constant once it reached 25 MPa, which was the preset value. Then, the CCD camera and the high-speed camera were used simultaneously to capture the speckle images on the surface of the specimens as reference images for the digital speckle correlation method. Finally, in the displacement control-loading mode, the specimens were loaded horizontally at the designated loading rate. In addition, the CCD camera was used to collect the speckle images on the surface of the specimens throughout the process, and the high-speed camera was activated by the post-trigger mode to acquire the speckle images on the surface of the specimens during each fault stick-slip motion until the end of the experiments.

## Results and discussion

### Characteristics of the stress evolution of the fault stick-slip at different loading rates

Fig 3 shows the typical loading curves for the six predefined loading rates. To better display the loading curves, only the first three stick-slip processes are presented. Table 1 shows the values of the stress drops corresponding to the results in Fig 3, where the marked points correspond to the analysis time of the displacement field. As shown in the Fig 3 and Table 1, the starting time of the first stick-slip of the specimens decreased as the loading rate increased, and the time it took for the stick-slip to occur decreased with increasing loading rate. The shear stress of the first stick-slip was 12.55–14.43 MPa, and the stress drop varied from 0.23 MPa to 1.14 MPa, indicating an insignificant correlation between the shear stress and the stress drop of the first stick-slip and the loading rate.

The shear stresses before and after the stick-slip were divided by the normal stress to obtain the static friction coefficient and the dynamic friction coefficient, respectively [18]. The curve illustrating the relationship between the loading rate and the static friction coefficient and the curve illustrating the correlation between the loading rate and the dynamic friction coefficient were plotted based on the averages of the static friction coefficients and the dynamic friction coefficients of the first three stick-slips at each loading rate (Fig 4). As can be seen from Fig 4, as the loading rate increased, the static friction coefficient fluctuated, implying an insignificant correlation between the loading rate and the static friction coefficient. Whereas, the dynamic friction coefficient increased as the loading rate increased. According to the above analysis, the slip characteristics during the inter-slip periods were more susceptible to the friction surface, and compared with the slip characteristics during the inter-slip period, the slip characteristics during the stick-slip period were more affected by the loading rate.

### Characteristics of the displacement evolution of the fault stick-slip at different loading rates

The digital speckle correlation method was used to analyze the speckle image on the surface of the specimen collected by the CCD camera during the loading process, thereby obtaining the evolution of the displacement field of the fault stick-slip, the corresponding relationship between the slip displacement of the fault stick-slip, the loading curve, and the evolution characteristics of the slip rate during the fault stick-slip at different loading rates.

Based on the evolution of the displacement field of the fault stick-slip, the evolution characteristics of the displacement of the fault stick-slip were similar under different loading rates. For a single stick-slip, the displacement generally underwent a uniform evolution phase, a non-uniform evolution phase, and an overall slip phase. Taking the deformation evolution analysis of the specimen at a loading rate of 10 μm/s (Fig 1) as an example, the horizontal displacement contour map of the sliding block obtained using the digital speckle correlation method (the displacement direction was the same as the displacement loading direction) is shown in Fig 5, in which the displacement contour interval was set to 0.02 mm. The point in time shown in Fig 5A corresponds to the time of Marked Point 1 in Fig 3. When the shear stress was 5.97 MPa, the displacement contours of the sliding block were sparse, the displacement value gradually decreased from the loading end to the fixed end, and the maximum displacement value was approximately 0.11 mm. The point in time shown in Fig 5B is consistent with that of Marked Point 2 in Fig 3. When the shear stress was 9.69 MPa, the contours near the loading end were relatively dense and approximately parallel, the displacement value gradually decreased from the loading end to the fixed end, and the maximum displacement value

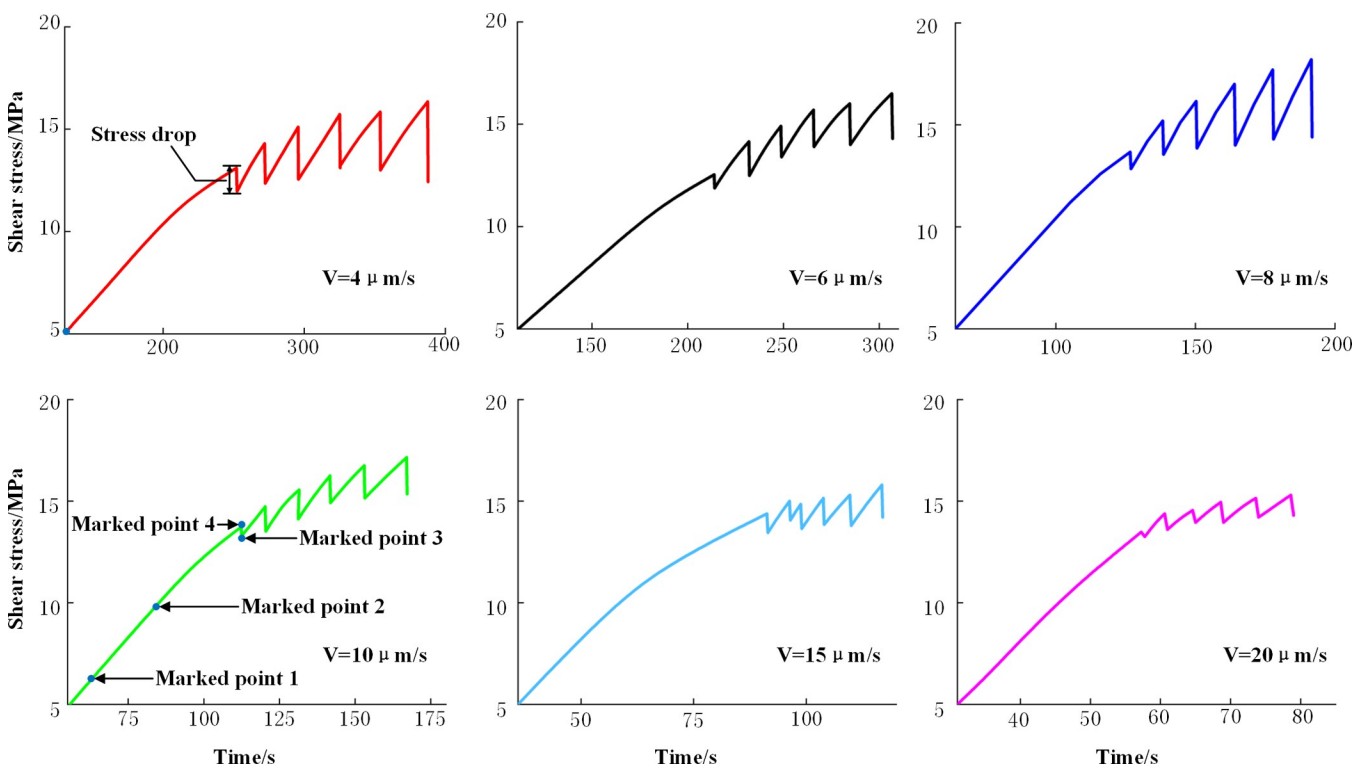

**Fig 3. Loading curves for the test.**

was approximately 0.24 mm, indicating that the fault displacement was in the uniform evolution phase. The point in time shown in Fig 5C corresponds to that of Marked Point 3 in Fig 3. When the shear stress was 13.62 MPa, the displacement of the sliding block at this moment increased significantly compared with that at the previous moment; the contours became denser, but the contours in different areas had different densities; and the maximum displacement value was approximately 0.71 mm, indicating a non-uniform displacement evolution. The point in time shown in Fig 5D is consistent with that of Marked Point 4 in Fig 3. When the fault stick-slip had already occurred and the shear stress was 13.19 MPa; the displacement of the sliding block experienced a mutation, but the intensity and distribution characteristics of the contours did not change significantly, and the maximum displacement was approximately 0.74 mm; indicating an overall slip phase.

The slip displacement of the fault stick-slip was calculated as follows: a 5 mm × 5 mm computation window 5 mm above and below the fault slip surface was selected, as shown in Fig 6. The average displacement of the speckles in the computation window represents the

**Table 1. Stress drop value.**

| Loading rate (μm/s) | 4 | 6 | 8 | 10 | 15 | 20 |
|---|---|---|---|---|---|---|
| Stress drop (MPa), 1st slip | 1.14 | 0.66 | 0.72 | 0.43 | 0.93 | 0.23 |
| Stress drop (MPa), 2nd slip | 1.95 | 1.66 | 1.65 | 1.21 | 0.94 | 0.78 |
| Stress drop (MPa), 3rd slip | 2.67 | 1.31 | 2.30 | 1.43 | 1.22 | 0.60 |
| Stress drop (MPa), 4th slip | 2.61 | 1.82 | 2.96 | 1.37 | 1.24 | 0.89 |
| Stress drop (MPa), 5th slip | 2.84 | 1.91 | 3.34 | 1.52 | 1.51 | 1.03 |
| Stress drop (MPa), 6th slip | 3.89 | 2.23 | 3.74 | 1.83 | 1.65 | 0.96 |

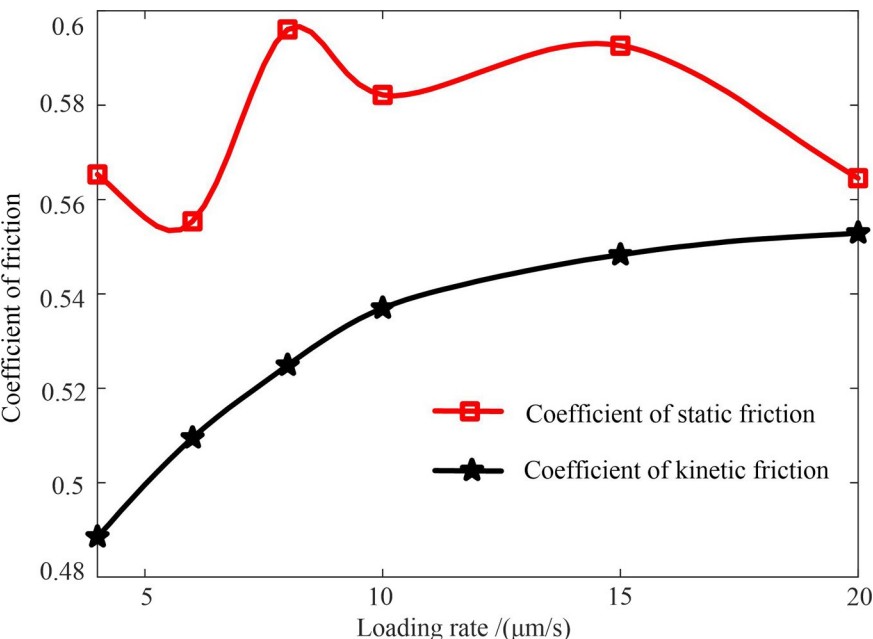

**Fig 4. Coefficients of static friction and coefficients of dynamic friction under different loading rates.**

displacement of the central point in the computation window, and the difference between the displacement components in the x-direction of the central point in the two computation windows denotes the slip displacement of the fault stick-slip.

According to the slip displacement of the fault stick-slips at different loading rates, for the same loading rate, there was a positive relationship between the slip displacement of the fault stick-slip and the stress drop, i.e., the stress drop increased with increasing sliding displacement. However, based on a comparison of the different loading rates, there was no relationship between the slip displacement and the stress drop. The evolution characteristics of the slip displacement of the fault stick-slip were analyzed at a loading rate of 4 μm/s. As can be seen from Fig 7, the sliding displacement of the rock interface was small during the period of slowly increasing shear stress (about 0–89 s). According to the analysis, this stage involved the compaction of the initial micro-bumps, which were randomly distributed by the sliding interface under normal stress. During the stage of approximately linear growth in shear stress (about 89–249 s), the interface sliding displacement increased non-linearly. According to the analysis, this stage was mainly caused by the non-uniform deformation of the sliding interface caused by the shear stress. During the stick-slip period, the interface sliding displacement suddenly increased, corresponding to a sudden drop in the shear stress. During the stick-slip period, the change in the interface sliding displacement was small, and the corresponding shear stress increase approximately linearly. Among them, the stress drop of the first stick-slip was 1.14 MPa, and the slip displacement was 0.10 mm. The stress drop of the second stick-slip was 1.95 MPa, and the slip displacement was 0.16 mm. The stress drop of the third stick-slip was 2.67 MPa, and the slip displacement was 0.27 mm. The stress drop of the fourth stick-slip was 3.03 MPa, and the slip displacement was 0.28 mm. The stress drop of the fifth stick-slip was 3.24 MPa, and the slip displacement was 0.31 mm. The stress drop of the sixth stick-slip was 3.91 MPa, and the slip displacement was 0.39 mm. For a given specimen, the increase in the stress drop occurred due to the sudden increase in the stick-slip displacement of the fault.

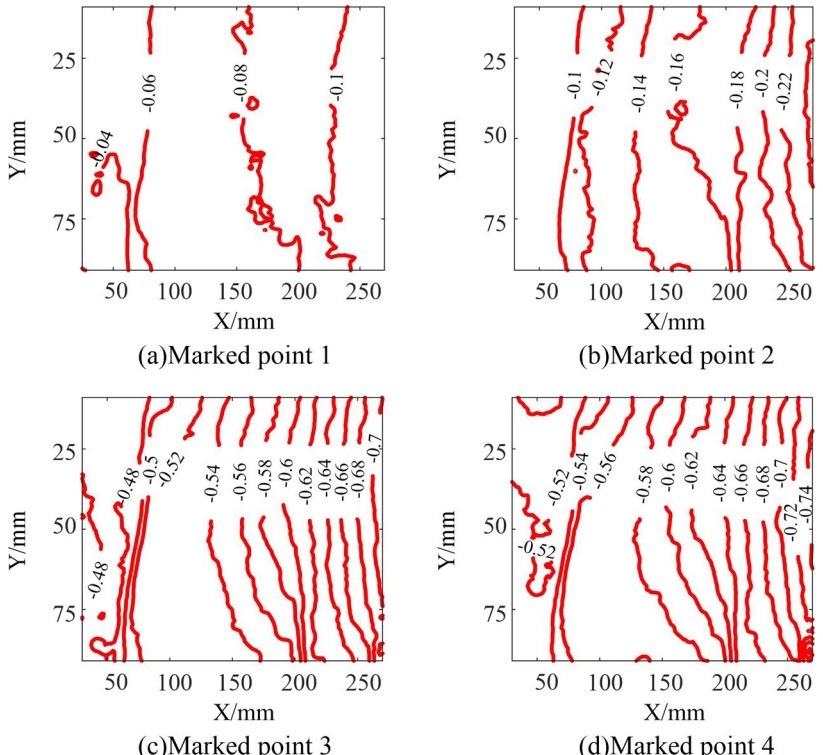

**Fig 5.** Displacement fields of the sliding blocks **(a)** Marked Point 1. (b) Marked Point 2. (c) Marked Point 3. (d) Marked Point 4.

The evolution of the slip rate of the fault during the stick-slip period indicates that the slip rate of the fault increased initially and then decreased; and as the number of stick-slips increased, the rate curve fluctuated and a significant multi-peak evolution occurred. There was no correlation between the slip rate, the stress curve, and the loading rate. Fig 8 shows the evolution of the slip rate of the fault during the stick-slip period at a loading rate of 4 μm/s. As can be seen from Fig 8, during the first stick-slip event and the second stick-slip event, the sliding rate increased rapidly in the initial stage of stick-slip, and then, it decreased to a stable state. In the third to sixth stick-slip events, the slip rate increased rapidly in the beginning of the slide, and then, it decreased. After this, the rate increased during the decline, reaching a new peak value of the rate, and then, it fluctuated around a stable state. Table 2 shows the increase in slip rate during the initial sticky slip from the first stick-slip to the sixth stick-slip event.

According to the above results of the evolution of the displacement field of the fault under different loading rates, the corresponding relationship between the slip displacement, the loading curve of the fault stick-slip, and the evolution characteristics of the slip rate of the fault during the stick-slip periods, the evolution of the displacement fields of the fault stick-slip was similar for different loading rates, showing three phases: uniform evolution, non-uniform evolution, and overall slip. Under different loading conditions, the slip displacement and the rate of the fault stick-slip were not correlated with the stress drop and loading rate of the loading curve.

## Magnitude and stress drop of fault stick-slip at different loading rates

Magnitude of the fault stick-slip: The magnitudes used in seismic studies include the Richter magnitude, the body wave magnitude, the surface wave magnitude, and the moment

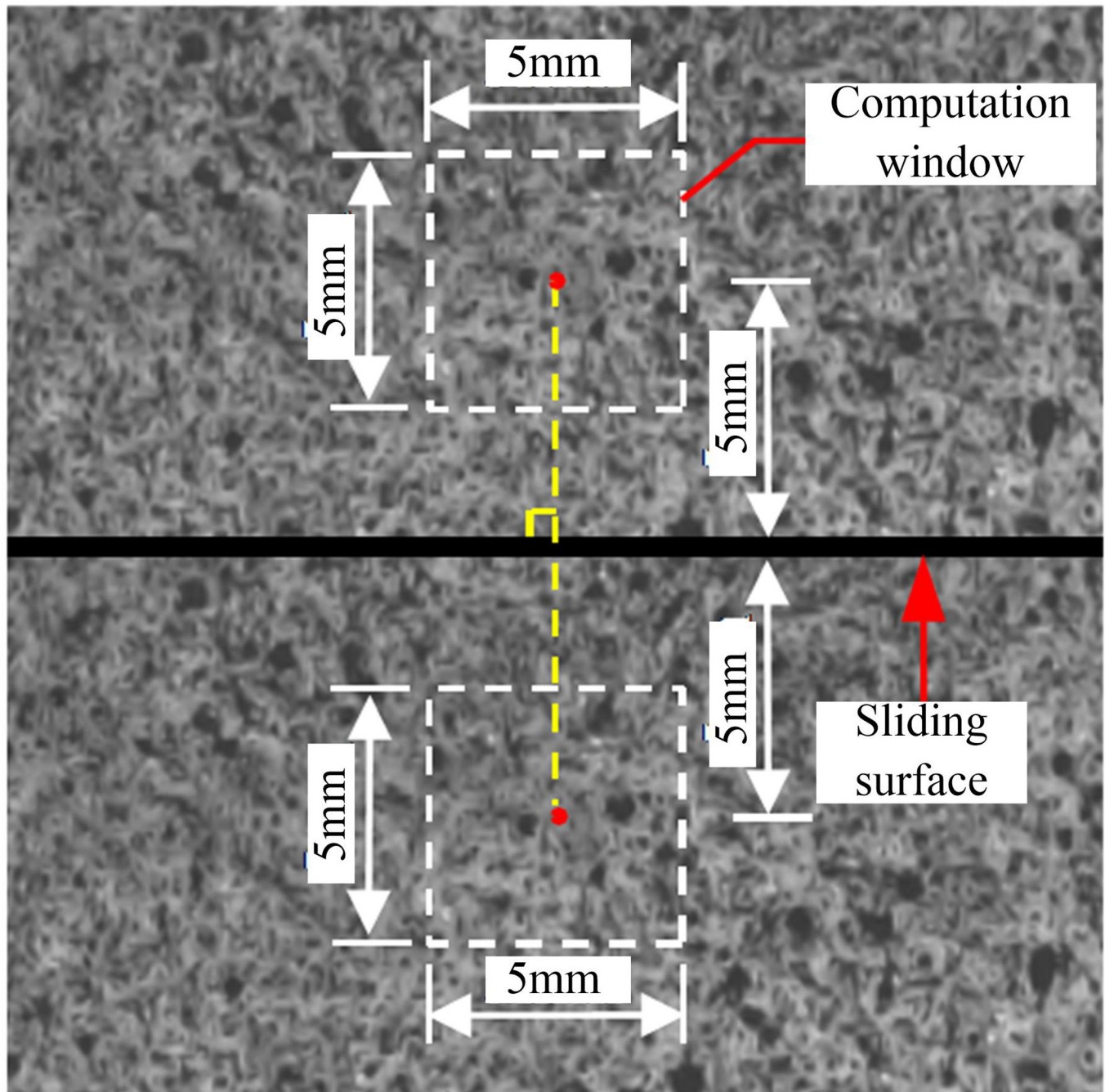

**Fig 6. Method for calculating the fault slip displacement.**

magnitude. In this study, the seismic moment M0 was used to analyze the moment magnitude of the fault stick-slip. The seismic moment M0 is a physical quantity characterizing the seismic intensity:

$$M_0 = GDA, \tag{1}$$

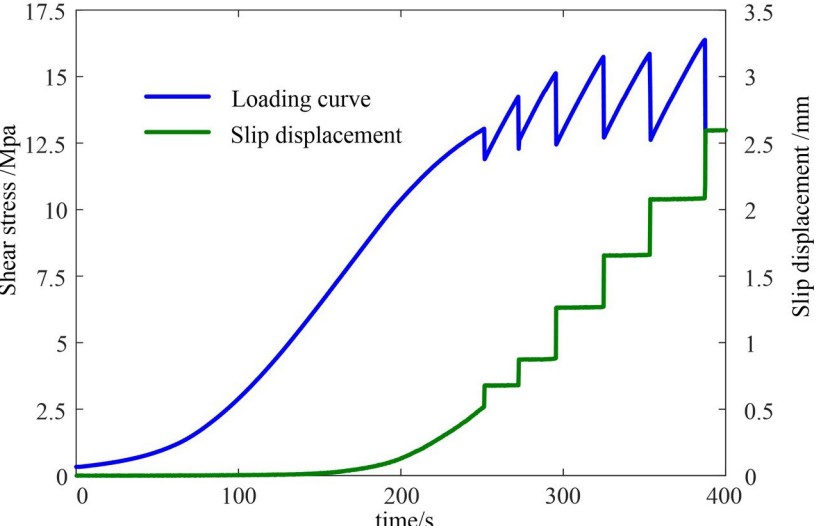

**Fig 7. The shear stress and fault slip displacement curves.**

where G is the shear modulus of the rock, D is the displacement of the fault stick-slip, and A is the area of the slip surface of the fault.

The seismic moment M0 was obtained using Eq (1), and then, the moment magnitude MW was calculated based on the empirical relationship between the seismic moment M0 and the moment magnitude MW [25,6]:

$$Mw = \frac{2}{3} \lg M_0 - 6.07. \tag{2}$$

Stress drop of the fault stick-slip: A fault stick-slip is a high-speed dynamic evolution process. Due to the low stress acquisition frequency of the experimental loading device commonly used, the actual stress drop in the fault stick-slip process cannot be accurately reflected. Consequently, the stress drop of the fault stick-slip process at the laboratory scale is several times higher than the on-site seismic stress drop, which leads to the illusion that the stress drop of small-scale earthquakes is higher than that of natural earthquakes. In the stress drop process, the speed change is not a simple linear decrease, but rather it is a complex deformation process. The speed evolution process includes multiple frequency components, the swing amplitude is large, and the speed high-frequency oscillation duration is short, but it is the main release of energy. During this stage, the strain and displacement also exhibit complex characteristics during the instability, and high-frequency oscillations occur. Since the stress drop determined in the experiments generally monotonically decreased, the actual stress drop was estimated based on the following equation [6]:

$$\Delta\sigma_V = \Delta\sigma \cdot \frac{t_V}{t}, \tag{3}$$

where $\Delta\sigma$ is the total stress drop, $t_V$ is the time of the sudden displacement during stick-slips, and t is the time of the actual stress drop.

Taking the third stress drop of the specimen at a loading rate of 20 μm/s as an example, the duration of the fault stick-slip acquired by the high-speed camera was 0.001 s, the time of the stress drop acquired by the test machine was 0.31 s, and the total stress drop was 0.60 MPa. Consequently, the actual stress drop corresponding to the specimen was $1.93 \times 10^{-3}$ MPa.

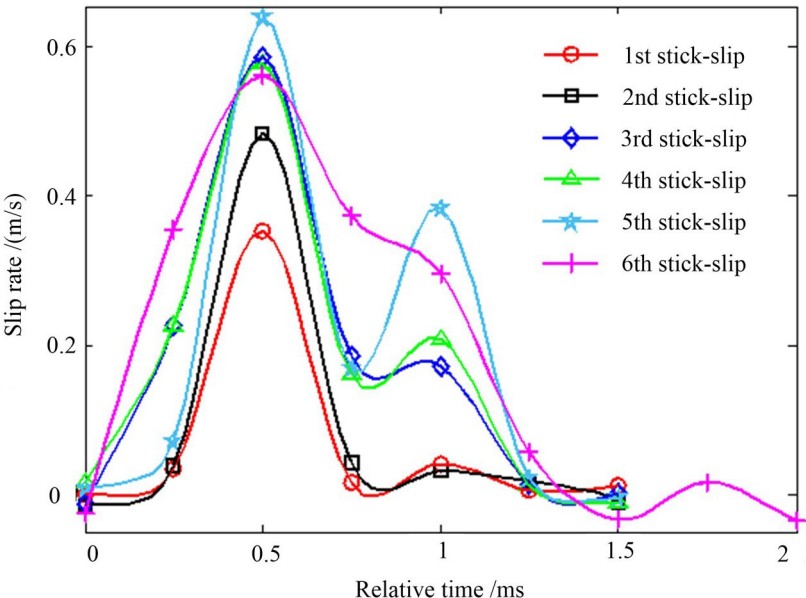

**Fig 8. The evolution curve of the slip rate during the fault stick-slip process.**

According to the above method, the effects of the loading rate on the magnitude and stress drop of the fault stick-slip were studied based on the experimental results obtained for the different loading rates.

Fig 9 shows the relationship between the magnitude and the actual stress drop under different loading rates. As can be seen from Fig 9, under the same vertical stress, the magnitude corresponding to the different loading rates is correlated with the actual stress drop. As the actual stress drop increased, the magnitude increased, indicating an approximately linear evolution. The magnitude obtained in the experiments ranged from −3.3 to −2.7.

Fig 10 shows the relationship between the magnitude, the actual stress drop, and the loading rate based on their average values. As the loading rate increased, the magnitude decreased; and as the loading rate increased, the actual stress drop decreased.

The loading rate is negatively correlated with the magnitude, and the actual stress drop is related to the rate-state dependent frictional constitutive relationship. [26,27,28] proposed a rate-state-dependent frictional constitutive relationship after a large number of experimental studies, which explains the friction and sliding instability mechanism of rocks well. The theory holds that friction is a function of the sliding rate and state variables, and the state variables represent the physical state of the contact surface. The sliding fault system has a critical stiffness determined by the friction parameters. When the effective elastic stiffness of the fault is less than the critical stiffness, the system will be unstable, i.e., an earthquake will occur. When the stiffness of the fault is slightly less than or close to the critical stiffness, the instability of the fault gradually becomes insignificant, and this is more characteristic of slow slip. That is, the loading rate and the state variables have an effect on the sliding characteristics of the fault.

**Table 2. Increase in slip rate in the initial stage of the stick-slip.**

| Times of stick slip | 1st slip | 2nd slip | 3rd slip | 4th slip | 5th slip | 6th slip |
|---|---|---|---|---|---|---|
| Slip rate(m/s) | 0.35 | 0.48 | 0.59 | 0.58 | 0.64 | 0.56 |

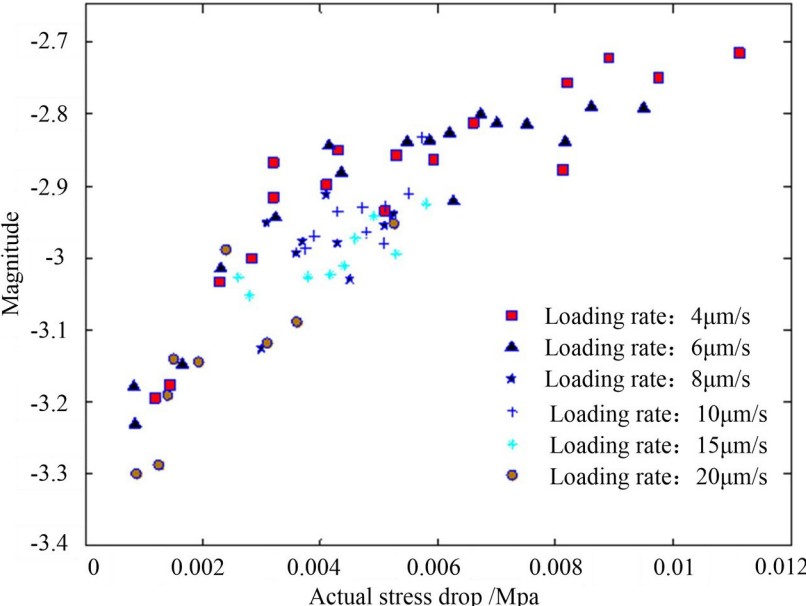

**Fig 9. The relationships between the magnitude and the actual stress drop at different loading rates.**

Based on the evolution results of the effects of the different loading rates on the stick-slip magnitude and stress drop of the fault in this experiment, even if the sliding properties of the fault have not changed, the magnitude and stress drop of the fault also the characteristics of stable sliding.

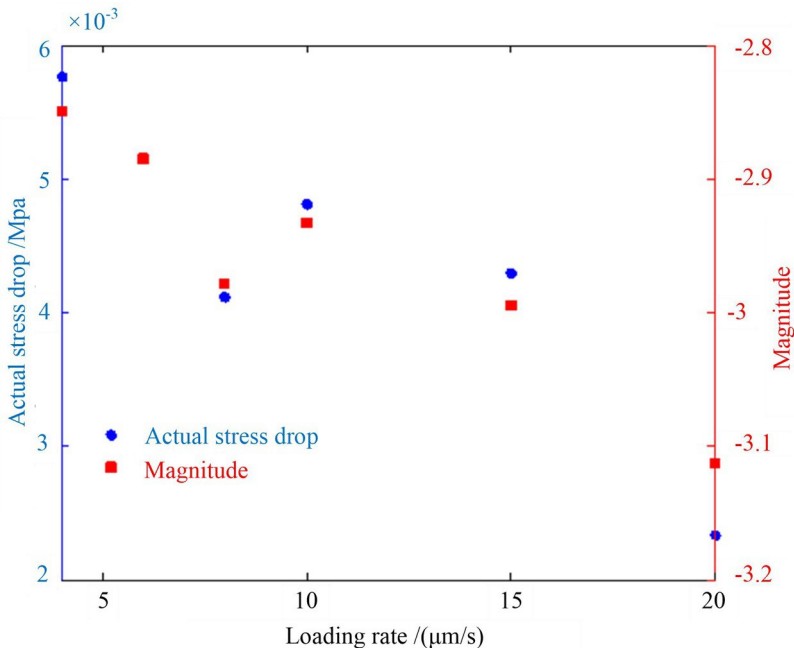

**Fig 10. Magnitude and actual stress drop responses to the loading rate.**

## Conclusions

In this experimental study, we simulated fault stick-slip using a double-sided shear experiment. The stress evolution and displacement evolution characteristics at different loading rates as well as the relationships between the stress drop, magnitude, and loading rate were obtained. The following main conclusions are drawn based on our results.

1. The evolutions of the displacement fields of the fault stick-slips were similar for different loading rates, i.e., the displacements exhibited three phases: uniform evolution, non-uniform evolution, and overall slip.

2. There was a correlation between the magnitudes corresponding to the different loading rates and the actual stress drop, which was generally exhibited as an approximately linear increase in magnitude with increasing actual stress drop.

3. The relationships between the magnitude and the actual stress drop and the loading rate can be generally expressed as follows: the magnitude decreased as the loading rate increased, whereas the actual stress drop decreased as the loading rate increased.

4. As the loading rate increased, although the sliding properties of the fault did not change, the magnitude and the stress drop of the fault also exhibited the characteristics of stable sliding.

## Acknowledgments

The authors would like to thank Letpub for the English language editing.

## Author Contributions

**Data curation:** Zong-xian Gao.

**Formal analysis:** Zong-xian Gao.

**Resources:** Xiu-hui Han.

**Writing – original draft:** Bao-xin Jia.

**Writing – review & editing:** Zong-xian Gao, Jia-xu Jin, Jian-jun Zhang.

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
