## [Decision Letter · Decision Letter 0]

24 Feb 2020

PONE-D-19-35929

Experimental Analysis of Stick-Slip Characteristics of Faults at Different Loading Rates

PLOS ONE

Dear Dr. Jia,

Thank you for submitting your manuscript to PLOS ONE. After careful consideration, we feel that it has merit but does not fully meet PLOS ONE’s publication criteria as it currently stands. Therefore, we invite you to submit a revised version of the manuscript that addresses the points raised during the review process.

We would appreciate receiving your revised manuscript by Apr 09 2020 11:59PM. To enhance the reproducibility of your results, we recommend that if applicable you deposit your laboratory protocols in protocols.io, where a protocol can be assigned its own identifier (DOI) such that it can be cited independently in the future. For instructions see: http://journals.plos.org/plosone/s/submission-guidelines#loc-laboratory-protocols

We look forward to receiving your revised manuscript.

Kind regards,

Peitao Wang

Academic Editor

PLOS ONE

Journal Requirements:

1. Please amend your Methods section to include the origin and vendor information and other characteristics of all materials and equipment used, in enough detail for another researcher to replicate the findings.

Reviewers' comments:

Reviewer's Responses to Questions

**Comments to the Author**

1. Is the manuscript technically sound, and do the data support the conclusions?

Reviewer #1: Yes

Reviewer #2: Yes

2. Has the statistical analysis been performed appropriately and rigorously? 

Reviewer #1: N/A

Reviewer #2: Yes

3. Have the authors made all data underlying the findings in their manuscript fully available?

Reviewer #1: Yes

Reviewer #2: Yes

4. Is the manuscript presented in an intelligible fashion and written in standard English?

Reviewer #1: No

Reviewer #2: Yes

5. Review Comments to the Author

Reviewer #1:

1) The authors conduct an interesting experimental study on stick-slip characteristic of faults at different load rates, which is extensively and intensively used in underground mine operations worldwide. As reviewer, I am very pleased to suggest the manuscript to be accepted for publication through aforementioned minor revisions. All comments and suggestions are as follows;

2) The English of the paper needs some editing. There are many typos in the paper

3) Introduction

The introduction is well articulated and the objective is well defined. However, the English needs to be edited

4) Materials and Methods

- The English needs to be edited

5) Please note, too many details are missed in this section. Why you choose the granite rock as the materials? It also didn’t give the interface roughness information which is important for evaluating the slide characters. The load equipment details, and why you choose such a series of load rates and a 25MPa vertical load? Moreover, give more details about the CCD and high speed camera.

6) Results and Discussion

7) The English needs to be edited, and you should explain what’s driving the phenomenon.

8) The results about the first stick-slip time varying the load rate are already well-known; they are not new. The reviewer suggests to the author to get more familiar with deformation and stress state in the interface which are responsible for the stick-slip character.

9) The results is lack of the mechanisms that can explain the phenomenon, the author should give more details.

Reviewer #2: This manuscript focuses on the fault stick-slip at different loading rates, and double-sided shear tests were conducted and the digital speckle correlation method were adopted for analysis. Besides the description of relevant experimental results, this article lacks theoretical analysis. Although the authors cited the empirical formulas of relevant scholars, they did not carry out an in-depth comparative analysis or not obtained new formulas containing the loading rate factor for reference. Therefore, this article also needs to be greatly improved and modified. Major revision is suggested. In addition, there are some details that need to be modified, as follows:

1)Some citations cannot be found in the references. Please check the full text carefully, for example: Guo et al.(2014).

2)More stick-slip stages are suggested to illustrate in Table 1 and Figure 3, only three stick-slip stages are not enough.

3)Fig. 9 shows the relationship between the magnitude and the actual stress drop under different loading rates, but there are so little points for some loading cases, such as 8,10,15μm/s. In addition, these data show that linear fitting may not be appropriate, please consider it.

4)Lines 221 to 227, please avoid this writing style, you can add a chart to explain it. Moreover, “For a given specimen, the slip displacement of the fault stick-slip increased as the stress drop increased.” In opinion of the reviewer, the stress drop should attribute to the sudden increase in displacement, not the opposite.

6. PLOS authors have the option to publish the peer review history of their article (what does this mean?). If published, this will include your full peer review and any attached files.

Reviewer #1: No

Reviewer #2: No

---

## [Author Response · Author response to Decision Letter 0]

13 Mar 2020

-Reviewer 1

The authors conduct an interesting experimental study on stick-slip characteristic of faults at different load rates, which is extensively and intensively used in underground mine operations worldwide. As reviewer, I am very pleased to suggest the manuscript to be accepted for publication through aforementioned minor revisions. All comments and suggestions are as follows;

1) The English of the paper needs some editing. There are many typos in the paper

 Authors’ response:

Thank you for your suggestion. It has been modified as required in the manuscript.

2) Introduction

The introduction is well articulated and the objective is well defined. However, the English needs to be edited

Authors’ response:

Thank you for your suggestion. It has been modified as required in the manuscript.

3) Materials and Methods

The English needs to be edited

Authors’ response:

Thank you for your suggestion. It has been modified as required in the manuscript.

4) Please note, too many details are missed in this section. Why you choose the granite rock as the materials? It also didn’t give the interface roughness information which is important for evaluating the slide characters. The load equipment details, and why you choose such a series of load rates and a 25MPa vertical load? Moreover, give more details about the CCD and high speed camera.

Authors’ response:

Thank you for your question.

(1)Granite is the most widely distributed in the stratum.The fault slip encountered in practical engineering generally occurs in the geological conditions of granite.It is universal to choose granite as experimental material.

(2)Under the condition of 25MPa vertical load, the roughness has little effect on the friction coefficient, and has great effect on the friction property. The main research object of this paper is the characteristics of fault stick slip. The main variable in the experiment is the loading rate. From the following calculation formulas,t can be known that the interface roughness is not a calculation parameter when calculating the magnitude of fault stick-slip and the stress drop.So it is not necessary to give the interface roughness.

(3)The load equipment details have been added in the Materials and Methods section. The content is as follows:“The experimental loading device was a RLJW-2000 hydraulic servo testing machine with a maximum axial load of 3000 kN and a maximum horizontal load of 1500 kN. The data recording frequency range of the loading device was 1×10−5 to 30 Hz; and the strain loading rate was 1×10−7 to 1×10−2 mm/s.”

(4)The higher the loading rate, the more stable the sliding.The lower the loading rate, the more unstable the sliding, and the more prone to stick-slip. Therefore, when the loading rate is small, the speed rate of each test increases less.

(5)When the loading rate is greater than 10, the speed rate of each test increases greatly.When the vertical pressure is less than 20 MPa, the surface roughness of the sample has a greater effect on the friction coefficient. When the vertical pressure is greater than 30 MPa, the sample will have some deformation. 25 MPa selected by comprehensive consideration.

(6)Details of CCD cameras and high-speed cameras have been added in the materials and methods section.The content is as follows:“CCD is the most commonly used image sensor for machine recording. In this experiment, the model of the CCD camera used was acA1600-20gm.”and “The high-speed camera was an industrial digital camera, which converted digital image capture targets into image signals and sent them to a dedicated image processing system. The image system performed various operations on these signals to extract the features of the targets. The high-speed camera model used in this experiment was a Photron Fast Cam SA1.1. ”

6) Results and Discussion

The English needs to be edited, and you should explain what’s driving the phenomenon.

Authors’ response:

Thank you for your suggestion. It has been added as required in the Conclusions.The content is as follows:“(4) As the loading rate increased, although the sliding properties of the fault did not change, the magnitude and the stress drop of the fault also exhibited the characteristics of stable sliding.”

8) The results about the first stick-slip time varying the load rate are already well-known; they are not new. The reviewer suggests to the author to get more familiar with deformation and stress state in the interface which are responsible for the stick-slip character.

Authors’ response:

Thank you for your suggestion. It has been modified as required in the Conclusions section of the manuscript.

9) The results is lack of the mechanisms that can explain the phenomenon, the author should give more details.

 Authors’ response:

Thank you for your suggestion. Theoretical analysis and interpretation were added to the Results and discussion.The content is as follows:lines 229 to 239,lines 254 to 259,lines 291 to 296,lines 322 to 335.

-Reviewer 2

This manuscript focuses on the fault stick-slip at different loading rates, and double-sided shear tests were conducted and the digital speckle correlation method were adopted for analysis. Besides the description of relevant experimental results, this article lacks theoretical analysis. Although the authors cited the empirical formulas of relevant scholars, they did not carry out an in-depth comparative analysis or not obtained new formulas containing the loading rate factor for reference. Therefore, this article also needs to be greatly improved and modified. Major revision is suggested. 

Authors’ response:

Thank you for your suggestion.Theoretical analysis and interpretation were added to the Results section of the manuscript.The content is as follows:lines 229 to 239,lines 254 to 259,lines 291 to 296,lines 322 to 335.

In addition, there are some details that need to be modified, as follows: Some citations cannot be found in the references. Please check the full text carefully, for example: Guo et al.(2014).

Authors’ response:

Thank you for your suggestion.We modified the format of the references.

2)More stick-slip stages are suggested to illustrate in Table 1 and Figure 3, only three stick-slip stages are not enough.

Authors’ response:

Thank you for your suggestion.We changed the original three stick-slip stages to six.

3)Fig. 9 shows the relationship between the magnitude and the actual stress drop under different loading rates, but there are so little points for some loading cases, such as 8,10,15μm/s. In addition, these data show that linear fitting may not be appropriate, please consider it.

Authors’ response:

Thank you for your suggestion.As the loading rate increases, the sliding will tend to be stable, and the number of stick-slips will decrease. So there are fewer points. The fitted lines for the points of 8, 10, and 15 μm / s are yellow, green, and red line segments, as shown in the following figure. We can find that the two fit well.

4)Lines 221 to 227, please avoid this writing style, you can add a chart to explain it. Moreover, “For a given specimen, the slip displacement of the fault stick-slip increased as the stress drop increased.” In opinion of the reviewer, the stress drop should attribute to the sudden increase in displacement, not the opposite.

Authors’ response:

Thank you for your suggestion.We changed “For a given specimen, the slip displacement of the fault stick-slip increased as the stress drop increased.” to “For a given specimen, the increase in the stress drop occurred due to the sudden increase in the slip-slip displacement of the fault.”

---

## [Decision Letter · Decision Letter 1]

25 Mar 2020

Experimental Analysis of the Stick-Slip Characteristics of Faults at Different Loading Rates

PONE-D-19-35929R1

Dear Dr. Jia,

We are pleased to inform you that your manuscript has been judged scientifically suitable for publication and will be formally accepted for publication once it complies with all outstanding technical requirements.

With kind regards,

Peitao Wang

Academic Editor

PLOS ONE

Additional Editor Comments (optional):

Reviewers' comments:

Reviewer's Responses to Questions

**Comments to the Author**

1. If the authors have adequately addressed your comments raised in a previous round of review and you feel that this manuscript is now acceptable for publication, you may indicate that here to bypass the “Comments to the Author” section, enter your conflict of interest statement in the “Confidential to Editor” section, and submit your "Accept" recommendation.

Reviewer #1: All comments have been addressed

Reviewer #2: All comments have been addressed

2. Is the manuscript technically sound, and do the data support the conclusions?

Reviewer #1: Yes

Reviewer #2: Yes

3. Has the statistical analysis been performed appropriately and rigorously? 

Reviewer #1: N/A

Reviewer #2: Yes

4. Have the authors made all data underlying the findings in their manuscript fully available?

Reviewer #1: Yes

Reviewer #2: Yes

5. Is the manuscript presented in an intelligible fashion and written in standard English?

Reviewer #1: Yes

Reviewer #2: Yes

6. Review Comments to the Author

Reviewer #1: The English of the paper have been reedited and meets the requirement of this journal, i agree that the editor can accept this paper.

Reviewer #2: It is a second round review. The authors reviesed the manuscript carefully and the quality of the paper is much improved. I suggest the manuscript be accepted for publication.

7. PLOS authors have the option to publish the peer review history of their article (what does this mean?). If published, this will include your full peer review and any attached files.

Reviewer #1: No

Reviewer #2: No

---

## [Editor Report · Acceptance letter]

2 Apr 2020

PONE-D-19-35929R1 

Experimental Analysis of the Stick-Slip Characteristics of Faults at Different Loading Rates 

Dear Dr. Jia:

I am pleased to inform you that your manuscript has been deemed suitable for publication in PLOS ONE. Congratulations! Your manuscript is now with our production department. 

With kind regards,

on behalf of

Dr. Peitao Wang 

Academic Editor

PLOS ONE